# Assessing the Perception and Contribution of Mangrove Ecosystem Services to the Well-Being of Coastal Communities of Chwaka and Menai Bays, Zanzibar

**Mohamed Khalfan Mohamed** [1,*], **Elhadi Adam** [1] **and Colbert M. Jackson** [1,2]

1   School of Geography, Archaeology and Environmental Studies, University of the Witwatersrand, Johannesburg 2050, South Africa; elhadi.adam@wits.ac.za (E.A.); colbermut@gmail.com (C.M.J.)

2   Department of Geography, Faculty of Natural and Agricultural Sciences, University of the Free State, Bloemfontein 9300, South Africa

*   Correspondence: mohamed.khalfan76@gmail.com or mohamed.khalfan@sumait.ac.tz; Tel.: +255-773-999-897

**Abstract:** The mangroves in Zanzibar are crucial to the survival of the local population, as they provide essential ecosystem goods and services. However, the actual value of mangrove products is not easily recognized. As a result, it is chiefly concluded that mangrove forests should be converted to uses that generate directly marketable products. This research sought to assess the perception and value of mangrove ecosystem services to the local communities around the Chwaka and Menai Bays. Key informant interviews, focus group discussions, and household surveys were used to collect data. The chi-squared test and one-way ANOVA were used to compare the awareness and perception of mangrove ecosystem services, respectively. The results show that provisioning services were the mangrove ecosystem services most identified by the household surveys, i.e., c. 84%. Supporting, regulating, and cultural services were rated in that order by 46.2%, 45.4%, and 21.0% of the respondents, respectively. This study found that there were statistically significant differences between Chwaka, Charawe, Ukongoroni, Unguja Ukuu, and Uzi wards in terms of households' awareness of regulating services ($\chi^2 = 6.061$, $p = 0.014$) and supporting services ($\chi^2 = 6.006$, $p = 0.014$). There were no significant differences in the identification of provisioning ($\chi^2 = 1.510$, $p = 0.919$) and cultural ($\chi^2 = 1.601$, $p = 0.901$) services. Occupations did not determine the reliance on mangrove ecosystem services ($\chi^2 = 8.015$; $p = 0.1554$). The approach used in this study can provide policymakers and land planners with a framework for the sustainable management of the ecosystem services provided by mangroves.

**Keywords:** mangrove ecosystem services; focus group discussions; key informant interviews; household survey; chi-squared test; Tukey's test; Zanzibar



## 1. Introduction

The concept of ecosystem services was first coined by King [1] and Helliwell [2], referring to how nature helps to support human societies. Afterward, Pearce [3], Pearce and Moran [4], Daily [5], Costanza and Folke [6], De Groot et al. [7], MEA [8], and Afonso et al. [9] were also interested in the concept of ecosystem services. The commonly recognized definition of ecosystem services states that they are the benefits that ecosystems provide to humans, which contribute to making human life both possible and worth living [8,10,11]. Ecosystem services are not a luxury but a crucial necessity for the well-being of humanity [8]. Ecosystem benefits are vital to our survival, providing us with the clean air, freshwater, fertile soil, and biodiversity that we depend on for food, raw materials for medicine, buildings, and industry [8,10,11]. Ecosystem services can be divided into four classes, i.e., provisioning, regulating, supporting, and cultural services [8,12,13]. Cultural services relate to the non-material world; they consist of activities like relaxation, art, thought, and religion that are difficult to measure in monetary terms [8]. Regulating

services include those relating to the capacity of natural and semi-natural ecosystems to control essential ecological processes and life support systems through biogeochemical cycles and other biosphere processes [8]. Provisioning services include ecosystem products and services provided for human consumption, such as food, raw materials, and energy [8]. Supporting services refer to natural processes such as cleaning the air and water, climate regulation and carbon sequestration, and the cycling of nutrients between organisms and the soil [8]. Supporting ecosystem services are essential for the production of all other ecosystem services.

The intensification of human activities has led to various environmental problems, putting more pressure on ecosystem services [14]. The degradation of ecosystem functions is an undeniable global challenge; it has attracted the attention of international sustainable development research [15]. The management of ecosystems services is a persistent challenge that demands effective solutions across diverse levels of decision-making. Without their proper management, humans risk losing the vital services provided by ecosystems [16]. Therefore, the adequate management of ecosystems is critical to their continued existence and sustainability [16]. A deep understanding of each ecosystem's unique characteristics and the stakeholders' diverse interests is crucial [16]. Thus, evaluating ecosystem services helps decision-making by highlighting the environmental functions gained and lost through exploitation and development [17].

Mangroves are distinctive halophytic plants that grow at the intersection of terrestrial, estuarine, and marine systems in subtropical and tropical coastal regions of about one hundred and twenty-three countries [18–20]. Mangroves provide various ecosystem services [21,22], e.g., wood, water, fuel wood, charcoal, paper, medicines, honey, and fodder [8,9]. Mangroves also provide recreational, educational, and research opportunities worth at least USD 1.6 billion annually [3]. Examples of mangrove-regulating services include mitigating the effects of climate change, such as protecting the coast from storms, flood, and erosion, and promoting biodiversity growth [8,9]. Mangrove forests contribute significant regulatory services, costing roughly 36,100 USD/ha [23]. Mangroves play a substantial role in regulating the climate by trapping carbon in soil and forest biomass. Mangroves provide around USD 6.7 billion in benefits, most of which come from carbon sequestration [24]. Also, mangroves exchange carbon dioxide with and emit methane into the atmosphere. Other regulating services include maintaining water quality in coastal areas by preventing saltwater intrusion. Examples of supporting services include habitats for breeding and spawning and commercial shrimp and fish farming [8,9].

Mangroves are in a grave decline despite international legislation protecting them due to rapid population growth and urbanization [25,26]. Over the past few decades, excessive exploitation of mangroves has severely diminished and fragmented mangrove ecosystems [21], eventually leading to a significant loss in terms of biodiversity [27]. The need for food, safe water, wood, fiber, and firewood, which has been rapidly rising, is primarily to blame; this has proven to be problematic for communities whose livelihoods largely depend on mangrove ecosystems [8]. Mangrove forests are currently disappearing at rates that are faster than those of tropical rainforests [20]. According to Hamilton and Casey [28] and Alam et al. [29], the growth of shrimp farms is responsible for up to 38% of global mangrove deforestation. Ecologists are concerned about the destruction and disappearance of mangroves due to the expansion of agriculture due to rapid population growth in coastal areas [30]. In addition, due to accelerated coastal development during the past 20 years, an annual decrease of between 0.16 and 0.39% of mangroves has been reported worldwide [31]. Severe damage to and fragmentation of mangroves impairs their ability to provide ecosystem services in the future [26,32].

Zanzibar's population of 1.1 million is marine-environment-reliant; the marine environment generates about 30% of Zanzibar's gross domestic product [33]. The gross domestic product per person in Zanzibar increased marginally to USD 1208 in 2022 from the previous year's 1099; approximately 50.3% of the population lives in poverty [34]. Zanzibar's mangroves are severely deteriorating due to unchecked tourism growth and related

infrastructural development [35]. Other factors include the rapid increase in population, poor fishing methods, excessive harvesting of mangrove products, disposal of untreated sewage from urban areas, and periodic coral bleaching [35]. Therefore, the unrestrained exploitation of mangroves has damaged the ecosystem and altered the significance of the ecosystem services that mangroves provide to the local communities, leading to increased poverty in Zanzibar. Because promoting human well-being, sustainability, and distributive justice is a significant requirement regarding harnessing mangrove ecosystem services, understanding the value of mangrove ecosystem services is essential [36].

Several studies, e.g., Quinn et al. [37], Kukkonen and Käyhkö [38], Käyhkö et al. [39], Nicholson [40], Othman [41], Lugomela [42], Mchenga and Ali [43], Mohamed et al. [44], and Mohamed et al. [45], have been conducted in Zanzibar to study mangrove ecosystems. However, these studies have documented very limited or no information regarding analyzing the contribution of mangrove ecosystem services in Zanzibar. In this regard, the present study aimed to assess the perception and value of mangrove ecosystem services to those living in the Chwaka and Menai Bay areas. This study specifically responded to the following queries: (1) Which mangrove ecosystem services exist in the Chwaka and Menai Bays? (2) What is the significance of the mangrove ecosystem services to the well-being of the Chwaka and Menai Bay communities? (3) Does the extraction of mangrove ecosystem services contribute to the destruction of mangrove forests? And (4) what are the factors affecting the supply of mangrove ecosystem services and how have they affected the livelihoods of the bay-adjacent communities?

## 2. Methodology and Study Locations

### 2.1. Study Locations

The study sites were the Chwaka and Menai Bays, the two largest mangrove bays on Unguja Island in Zanzibar (Figure 1). In 1997, the Revolutionary Government of Zanzibar declared both sites protected zones. Ten mangrove species exist on the mainland and Zanzibar [42]. Chwaka Bay has the largest mangrove area of c. 2294 ha [42]. All ten species of mangrove in Tanzania are present in the forests of Chwaka Bay [46]. Menai Bay is composed of c. 988 ha of mangroves. Because of extensive deforestation, Menai Bay's ecological makeup differs significantly from the nearby Jozani mangrove forest. The Charawe, Chwaka, Ukongoroni, Unguja Ukuu, and Uzi wards occupy the Chwaka and Menai Bay areas, and according to the 2022 census report, they have human populations of 954, 3196, 896, 1563, and 1801, respectively. These communities rely on the mangrove ecosystems in the two bays for their survival. Zanzibar, which was designated a UNESCO World Heritage Site in 2000, has exceptional beaches, coral reefs, and a rich culture [8].

### 2.2. Data Collection

This research employed a mixed-methods design that combined qualitative and quantitative research elements. Fieldwork was conducted from February to May 2022. Household surveys, focus group discussions, and key informant interviews were used to collect data. Respondents were informed of the study's goals and signed an informed consent form. The focus group discussion approach elicited the participants' beliefs, perceptions, attitudes, and experiences toward their interaction with the mangrove ecosystem services in the study areas [14]. In focus group discussions, data can be obtained from a purposely chosen group of individuals and not from a statistically representative sample of a broader population [47]. The focus group discussions were held in the wards of Chwaka, Uzi Ukongoroni, Unguja Ukuu, and Charawe. The focus group discussions comprised Shehas (local administrators), villagers, herbalists, active/retired fishermen, former/present mangrove harvesters, and vendors of fish products. Others were members of the Jozani-Chwaka Bay Community Organization and ward-level forest officers.

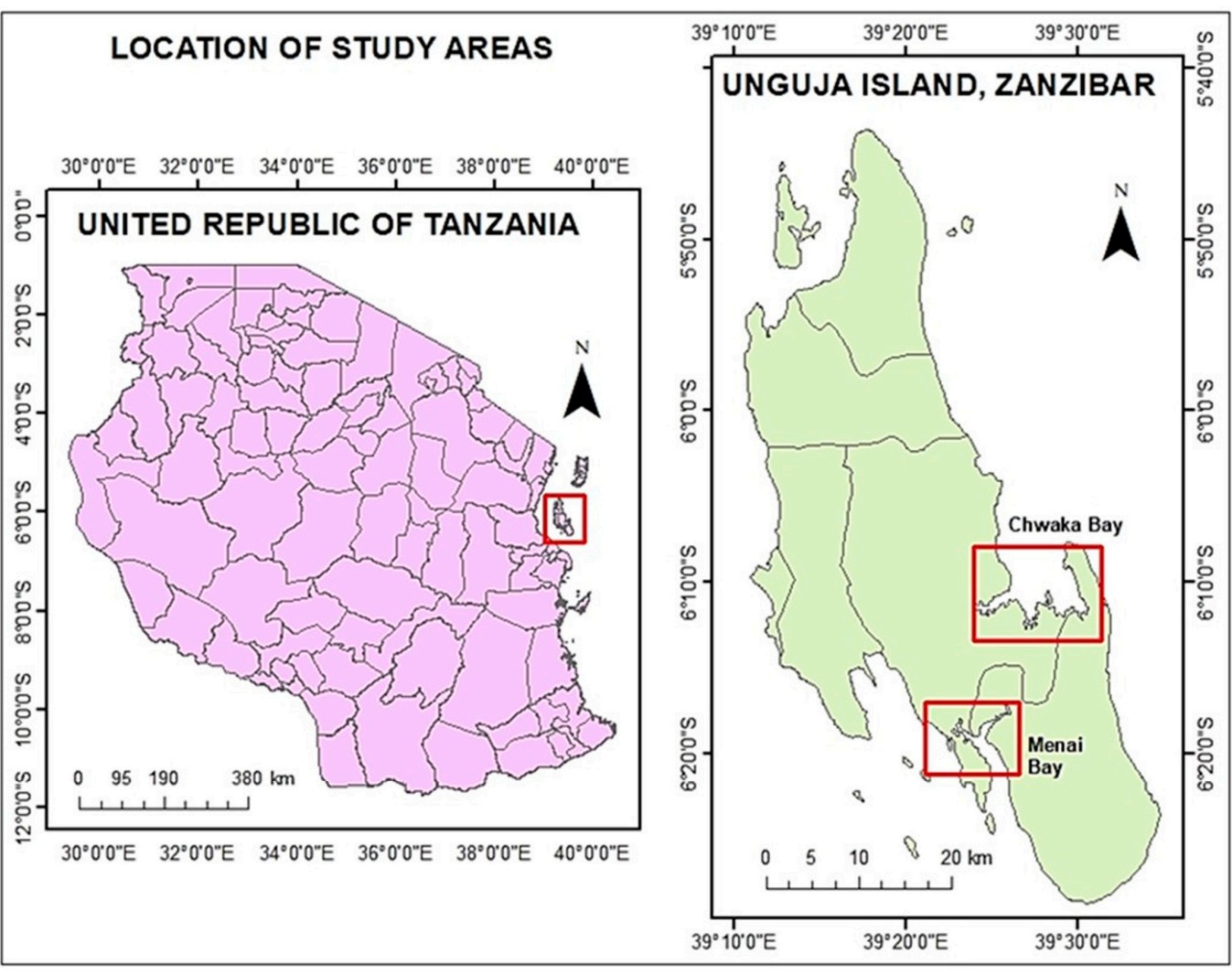

**Figure 1.** Location of Chwaka and Menai Bays in Unguja, Zanzibar.

Additionally, in the key informant interviews, the members were government officers from the Ministry of Agriculture and Natural Resources, Zanzibar's House of Representatives, the regional commissioner's office, the district office, and academics from universities in Zanzibar who held expertise in mangrove ecosystems. Government representatives were also interviewed. Academic staff members were from Abdulrahman Al-Sumait University, Zanzibar University, the Institute of Marine Science, and the State University of Zanzibar. The topics chosen for the semi-structured interviews were as follows:

— The concept of ecosystem services;
— Ecosystem services of mangroves;
— Categorization of mangrove ecosystem services;
— Perceptions toward mangrove ecosystem services;
— The functioning of mangrove ecosystems;
— Anthropogenic/natural threats to mangroves; and
— Mangrove socio-ecology and management.

Also, a household survey was conducted using a structured questionnaire (Appendix A). Structured questionnaires are typically viewed as the best choice for large respondent populations and when detailed information is required. Questionnaires were administered to heads and/or adult family members of households. The questionnaire was updated using responses gathered from the focus group discussions and key informant interviews.

*2.3. Sample Size and Sampling Design*

For the focus group discussions, snowballing sampling was used in this study to identify participants knowledgeable about mangrove ecosystem services and/or who resided in the Chwaka and Menai Bay areas. Only five focus group discussions were held; the groups comprised from 9 to 11 members. The selection criteria for the key informants included age (generally 40 years and older), expertise in mangrove ecosystems, willingness to participate, familiarity with mangrove ecosystem services in the study area, and whether they were born and raised in the respective wards. Other factors included the influence of the participants in the decision-making process in sustainable mangrove management and whether the informants were beneficiaries (either directly or indirectly) of mangrove ecosystem services in the study area. For the key informant interviews, purposeful sampling was applied. In total, 31 key informants were interviewed. According to the 2022 Population and Housing Census, Chwaka, Ukongorono, Charawe, Unguja Ukuu, and Uzi had 317, 179, 181, 160, and 174 households, respectively. In this regard, 10% per household, i.e., 102 households, were chosen to participate in the household survey. The multistage sampling technique was used to select the respondents. First, villages were selected from within the wards in the study area, homesteads were randomly selected, and finally households were selected. In order to test the research instruments, 10% of the sample was used in a pilot study [48].

*2.4. Data Analysis*

This study used the SPSS statistical software version 23 (IBM Corp., Armonk, NY, USA) to conduct the statistical analysis. For comparison purposes, three groups were created by merging some of the initial five wards. Chwaka ward formed the first group because it had a substantially higher number of participants than the other four wards, i.e., 32. Ukongorono and Charawe wards have similar social and physical characteristics and are close to each other; therefore, they were merged to form a second group, i.e., 18 + 18 = 36. Unguja Ukuu and Uzi wards were combined to create a third group since they also share the same physical and social characteristics, i.e., 16 + 18 = 34. Awareness of mangrove ecosystem services was compared using a chi-squared test. In contrast, one-way ANOVA was used to compare the perception of the respondents on the relative importance of mangrove ecosystem services between the wards. The relationship between the significance of the perceived mangrove ecosystem services and the occupation of the participants was determined using an a posteriori multiple comparison test (Tukey's test).

The communities in the study area depend on mangrove ecosystems for their livelihoods [14]; therefore, their socio-economic status and government forestry policy determine how they perceive mangrove ecosystem services. To determine how the variables affected the awareness of ecosystem services of the community, the respondents were classified based on (i) age, (ii) gender, (iii) marital status, (iv) level of education, (vii) household income, (vi) household size, (vi) occupation, (vii) accessibility to mangrove forest, (viii) how long the household has been in the area, and (ix) government forestry policy (Table 1). The Likert scale was used to rate responses [49].

The results from the focus group discussions and key informant interviews were examined and processed using two qualitative data analysis methodologies. Firstly, constant comparison analysis was used, i.e., the data were grouped into small units, and then codes were attached to each of the units. Secondly, the codes were grouped into categories. Finally, themes expressing the content of each of the groups were developed [50,51]. Discourse analysis was applied to understand the language used by the members of the focus group discussions and key informant interviews [47].

**Table 1.** Variables used in the logistic regression analysis.

| Variable | Description of Variables |
|---|---|
| Gender | Either male or female |
| Age | Age of interviewees |
| Marital status | Single, married, divorced, etc. |
| Level of education | Informal, primary, secondary, post-secondary, etc. |
| Income of households | Average monthly income of household |
| Size of households | Number of family members in household |
| Occupation | Subsistence farmers, fishermen, public servant, small scale business, education, crafts, etc. |
| Accessibility to mangroves | Full access or access denied for harvesting mangroves |
| Household residence period | The number of years the household has lived in the area |
| Government policy | Community-based forest management or not |

## 3. Results

### 3.1. Demographics and Socioeconomics of the Respondents of the Household Survey

Overall, Muslims dominated the respondents to the household survey, i.e., 99.4%, while Christians comprised 0.6% (Table 2). Male participants comprised 55.7%, while the remainder were females. About 49%, 53%, and 60% of the participants in the Chwaka, Charawe/Ukongoroni, and Unguja Ukuu/Uzi wards were in the 50 to 59 age range, respectively. Overall, 87% of the participants were married. Only 4.9% of the respondents in the household survey had attained tertiary education. On average, about 62% of the households had from 6 to 10 family members. About 68% of the homes in the wards used mangroves as their primary fuel source.

**Table 2.** Demographic variables of participants in the household survey in Chwaka and Menai Bays, Unguja, Zanzibar.

| Variable | Sub-Category | Chwaka Ward (%) | Charawe/Ukongoroni Wards (%) | Unguja Ukuu/Uzi Wards (%) | Average |
|---|---|---|---|---|---|
| Religion | Muslims | 99.1 | 99.4 | 99.6 | 99.4 |
| | Christians | 0.9 | 0.6 | 0.4 | 0.6 |
| Gender | Male | 61.1 | 49.1 | 56.9 | 55.7 |
| | Female | 38.9 | 50.9 | 43.1 | 44.3 |
| Age | 40–49 | 8.7 | 32.1 | 23.4 | 21.4 |
| | 50–59 | 48.8 | 52.6 | 59.7 | 53.7 |
| | 60+ | 42.5 | 15.3 | 16.9 | 24.9 |
| Marital status | Married | 88.2 | 81.7 | 93.1 | 87.7 |
| | Widowed | 9.0 | 4.6 | 5.2 | 6.3 |
| | Divorced | 2.8 | 13.7 | 1.7 | 6.1 |
| Education level | Tertiary | 4.1 | 3.4 | 7.3 | 4.9 |
| | Secondary | 41.5 | 31.8 | 38.1 | 37.1 |
| | Primary | 54.4 | 64.8 | 54.6 | 57.9 |
| Household size | 1–5 | 11 | 21.3 | 49.2 | 27.2 |
| | 6–10 | 69.8 | 66.8 | 48.7 | 61.8 |
| | 10+ | 19.2 | 11.9 | 2.1 | 11.1 |

The household survey showed that 41.3% of the participants were fishermen, 35.8% were subsistence farmers, and only 0.1% of the participants were beekeepers (Figure 2a). About 27% of the respondents earned >200,000 TZS (1 USD, 2504.76 TZS). The respondents who had a mean monthly income of about 100,000 to 200,000 TZS comprised 62.3% (Figure 2b).

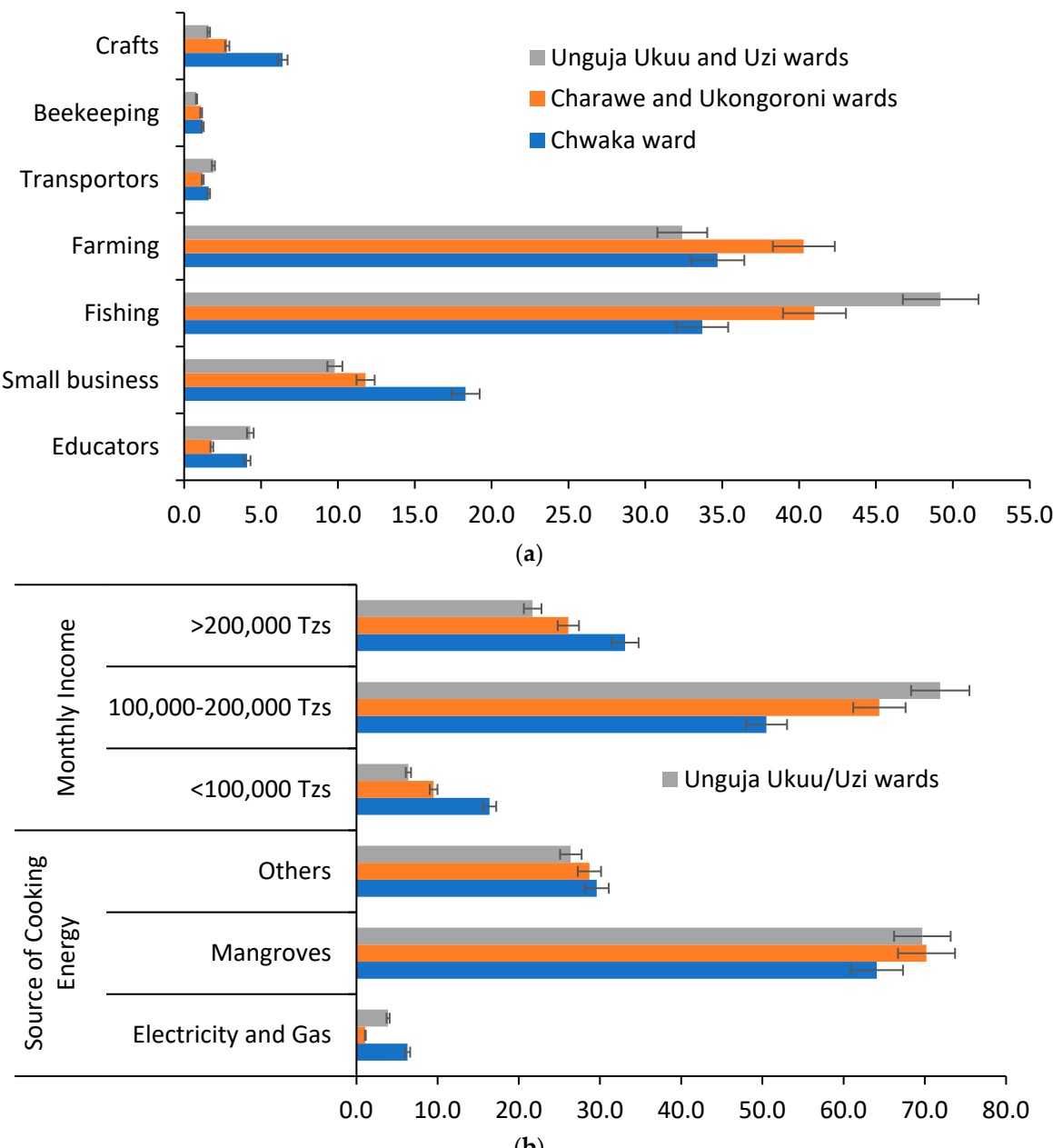

**Figure 2.** (**a**) Occupation of respondents (in percentage) in Chwaka, Ukongoroni, Charawe, Unguja Ukuu, and Uzi wards in Chwaka and Menai Bays, Unguja Island, Zanzibar. Error bars are 95% confidence intervals. (**b**) Average monthly income and source of cooking energy of the households in Chwaka, Ukongoroni, Charawe, Unguja Ukuu, and Uzi wards in Chwaka and Menai Bays, Unguja Island, Zanzibar. Error bars are 95% confidence intervals.

*3.2. Classification of Mangrove Ecosystem Services*

The focus group discussions and key informant interviews identified sixteen mangrove ecosystem services in the study area (Table 3), categorized into four classes, i.e., supporting, cultural, provisioning, and regulating services. Two supporting services (wild habitat and soil formation), four cultural services (ecotourism, spiritual activities, educational purposes, and scenic viewing), seven provisioning services (poles, firewood, medicine, honey, fruit, picnic sites, and fodder), and three regulating services (erosion control and sediment accretion, regulation of climate, and attenuation of waves) were identified by the respondents.

**Table 3.** Mangrove ecosystem services identified in the study area through focus group discussions and key informant interviews.

| Mangrove Ecosystem Service | Use of Mangrove Ecosystem Services Identified in the Study Area |
|---|---|
| Wild habitat (SS) | Home to numerous shrimp, mollusks, fish, and crabs that serve as food sources. |
| Soil formation (SS) | Sediments are filtered by mangroves into soil; this soil is used to make bricks. |
| Ecotourism (CS) | Mangroves are popular tourist destinations because of their biodiversity. |
| Spiritual activities (CS) | Mangrove forests are used for ritual practices, e.g., to pray for rain and household protection, and they also contain ritual plants used for ritual healing. |
| Education (CS) | Source of information and knowledge regarding mangroves. |
| Scenic viewing (CS) | Viewing of the natural scenery of mangroves so as to feel relaxed and refreshed. |
| Poles (PS) | For building houses and boats and making furniture and fences, particularly *Rhizophora mucronata*, *Bruguiera gymnorhiza*, and *Ceriops tagal*. |
| Firewood (PS) | *Rhizophora mucronata* and *Ceriops tagal* are a source of cooking energy. |
| Medicine (PS) | Leaves of *Xylocarpus granatum* are used to treat stomachache. |
| Honey (PS) | Honey is used to disinfect wounds and heal burns. It is also added to tea and porridge and spread on bread. |
| Fruits (PS) | The fruits of *Avicennia marina* are chewed to ease heartburn. |
| Picnic sites (PS) | Areas set aside inside mangrove forests where people can visit for outdoor activities |
| Fodder (PS) | The leaves of particularly *Heritiera littoralis* and *Avicennia marina* are fed to livestock. |
| Erosion control and sediment accretion (RS) | Minergenic mangroves trap and consolidate sediment; this controls soil erosion and leads to improved soil fertility for farming. |
| Regulation of climate (RS) | Mangroves sequester carbon in soils and forest biomass and exchange carbon dioxide with and emit methane to the atmosphere. |
| Attenuation of waves (RS) | Mangrove vegetation causes wave attenuation because it acts as an obstacle by creating vegetation drag for the water flow in the waves. |

*3.3. Awareness of Mangrove Ecosystem Services*

The findings indicated that the participants in the household survey were aware of the benefits given by mangroves (Figure 3). Overall, 83.9% of the households knew provisioning services to be the most crucial, followed by supporting, regulating, and cultural services with 46.2%, 45.4%, and 21.0%, respectively. Chwaka ward was more aware of provisioning services, followed by supporting services, regulating services, and, lastly, cultural services, as they attained 90.4%, 39.4%, 36.6%, and 25.2%, respectively. Ukongoroni and Charawe wards were more aware of provisioning services, i.e., 92.6%, followed by regulating services with 41.3%, supporting services with 37.4%, and cultural services with 28.1%. The findings on the awareness of mangrove ecosystem services from Unguja Ukuu and Uzi households indicated that provisioning services attained the highest score of 68.7%, followed by supporting, regulating, and cultural services, which earned 61.8%, 58.4%, and 9.8%, respectively. The respondents in the focus group discussions in Unguja Ukuu and Uzi wards reported that several had attended training on mangrove ecosystems, which made them well-versed in the supporting and regulating services of mangroves. This study found that there were statistically significant differences between the Chwaka, Charawe, Ukongoroni, Unguja Ukuu, and Uzi wards in terms of household awareness of regulating services ($\chi^2 = 6.061$, $p = 0.014$) and supporting services ($\chi^2 = 6.006$, $p = 0.014$). There were no substantial differences between the wards in the identification of provisioning ($\chi^2 = 1.510$, $p = 0.919$) and cultural ($\chi^2 = 1.601$, $p = 0.901$) services.

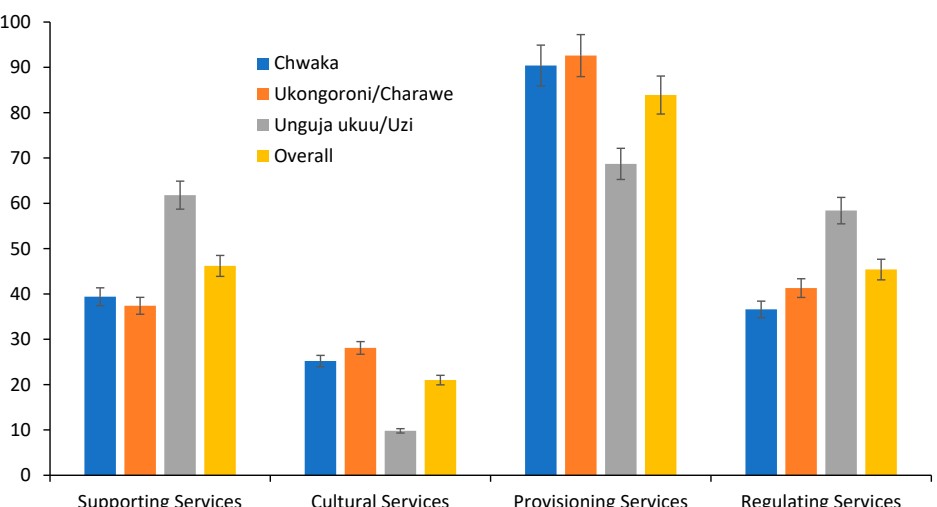

**Figure 3.** Awareness of mangrove ecosystem services in Chwaka, Charawe, Ukongoroni, Unguja Ukuu and Uzi wards in Unguja, Zanzibar. Error bars are 95% confidence intervals.

The following radar diagram analysis (Figure 4) shows that the residents of the Chwaka ward needed to be more aware of mangrove ecosystem services compared to those of the Ukingoroni, Charawe, Unguja Ukuu, and Uzi wards. The five mangrove ecosystem services that the Chwaka, Ukongoroni, and Charawe wards primarily identified were poles, firewood, medicine, spiritual activities, and educational purposes, as opposed to the Unguja Ukuu and Uzi wards' protection of shorelines, regulation of climate, wild habitats, soil formation, and sediment trapping. In contrast to supporting and regulating services, which were more recognized in the Chwaka, Ukongoroni and Charawe wards, the Unguja Ukuu and Uzi wards mostly recognized the mangrove ecosystem services in the form of cultural and provisioning services. The mangrove ecosystem services with the highest identification score were poles (91.4%), attenuation of waves (91.2%), and regulation of climate (89.6%), while fodder harvesting (9.4%) and spiritual activities (11.2%) were among the lowest-identified mangrove ecosystem services.

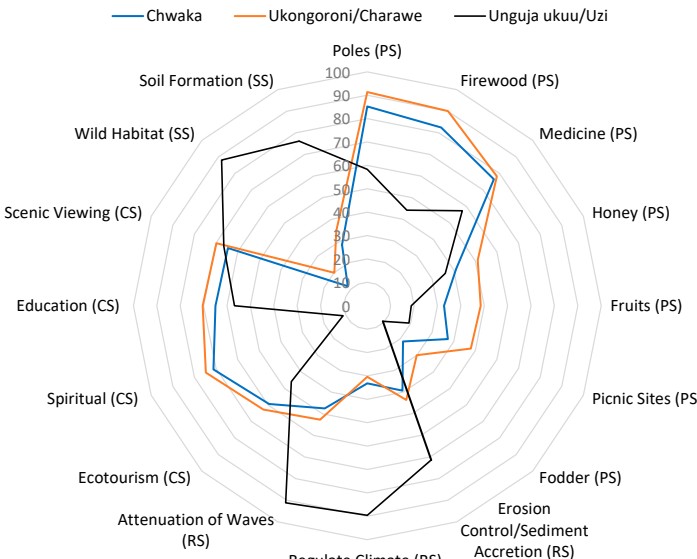

**Figure 4.** Awareness (in percentage) of the identified mangrove ecosystem services by households in the study areas: PS (provisioning services), RS (regulating services), CS (cultural services), and SS (supporting services).

### 3.4. Mangrove Ecosystem Services as Perceived by Household Survey

There were notable disparities in the perception of mangrove ecosystem services between the wards (Table 4). The Chwaka, Ukongoroni, and Charawe wards considered provisioning and cultural services to be more critical compared to the Unguja Ukuu and Uzi wards, where regulating and supporting services were more significant. The average scores of provisioning services were 1.6, 2.4, and 1.9 in the Unguja Ukuu/Uzi, Ukongoroni/Charawe, and Chwaka wards, respectively. The results for regulating services showed 3.5, 1.5, and 1.5 for the Unguja Ukuu and Uzi, Ukongoroni and Charawe, and Chwaka wards, respectively. The cultural services for the Unguja Ukuu and Uzi, Ukongoroni and Charawe, and Chwaka wards scored 1.8, 2.4, and 2.2, respectively. The household survey indicated that the Unguja Ukuu and Uzi, Ukongoroni and Charawe, and Chwaka wards attained mean scores of 3.3, 1.3 and 1.2, respectively.

**Table 4.** The importance of mangrove ecosystem services in the study area according to the household survey: PS (provisioning services), RS (regulating services), CS (cultural services), and SS (supporting services).

| Category | Mangrove Ecosystem Services | Chwaka Ward | Ukongoroni/Charawe Wards | Unguja Ukuu/Uzi Wards | Overall |
|---|---|---|---|---|---|
| PS | Poles | 3.4 (0.5) | 3.9 (0.3) | 2.4 (0.4) | 3.2 (0.4) |
| | Firewood | 2.5 (0.7) | 3.8 (0.5) | 1.5 (0.5) | 2.6 (0.6) |
| | Medicines | 2.5 (0.5) | 2.8 (0.7) | 2.3 (0.6) | 2.5 (0.6) |
| | Honey | 1.4 (0.6) | 1.9 (0.6) | 1.5 (0.3) | 1.6 (0.5) |
| | Fruit | 1.3 (0.6) | 1.7 (0.4) | 1.2 (0.3) | 1.4 (0.6) |
| | Picnic sites | 1.4 (0.7) | 1.6 (0.2) | 1.2 (0.4) | 1.4 (0.4) |
| | Fodder | 1.1 (0.4) | 1.3 (0.3) | 1.1 (0.2) | 1.2 (0.3) |
| | Ave. PA | 1.9 (0.6) | 2.4 (0.4) | 1.6 (0.3) | 1.9 (0.4) |
| RS | Erosion control and sediment accretion (RS) | 1.4 (0.6) | 1.5 (0.4) | 2.9 (0.6) | 1.9 (0.5) |
| | Regulation climate | 1.3 (0.5) | 1.2 (0.4) | 3.8 (0.4) | 2.1 (0.4) |
| | Attenuation of waves (RS) | 3.8 (0.6) | 2.0 (0.7) | 3.9 (0.3) | 3.2 (0.5) |
| | Ave, RS | 1.5 (0.6) | 1.5 (0.5) | 3.5 (0.4) | 2.2 (0.4) |
| CS | Ecotourism | 2.0 (0.6) | 2.2 (0.3) | 1.9 (0.6) | 2.0 (0.5) |
| | Spiritual | 2.3 (0.7) | 2.7 (0.8) | 1.1 (0.2) | 2.0 (0.6) |
| | Education | 2.2 (0.4) | 2.5 (0.3) | 1.9 (0.6) | 2.2 (0.4) |
| | Scenic viewing | 2.3 (0.5) | 2.5 (0.4) | 2.4 (0.3) | 2.4 (0.5) |
| | Ave. CS | 2.2 (0.6) | 2.4 (0.4) | 1.8 (0.4) | 2.1 (0.5) |
| SS | Wild habitat | 1.1 (0.4) | 1.2 (0.6) | 3.7 (0.4) | 2.0 (0.5) |
| | Soil formation | 1.2 (0.4) | 1.4 (0.3) | 2.9 (0.6) | 1.8 (0.4) |
| | Ave, SS | 1.2 (0.4) | 1.3 (0.4) | 3.3 (0.5) | 1.9 (0.4) |

Note: means are represented by the values outside the parentheses, while the values inside the parentheses represent standard deviations. Likert scale: not important (1), least important (2), moderately important (3), most important (4), and not sure (5).

Table 5 illustrates the perceptions of mangrove ecosystem services by occupation in the Chwaka, Ukongoroni, Charawe, Unguja Ukuu, and Uzi wards. The findings showed that mangrove ecosystem services were especially significant in some cases when they had a solid connection to occupation. For example, the importance of provisioning services, such as poles, was rated highly by small-scale farmers (3.6), servants in the public sector (3.5), small-scale businesses (3.5), and craftsmen/women (3.4) (Table 5). All occupations indicated less importance to honey. Regulating services such as soil erosion control and sediment accretion, regulation of climate, and attenuation of waves were highly rated by the educated and those in the teaching/research profession in particular. Cultural services, such as scenic viewing, were lowly perceived as a mangrove ecosystem service by subsistence farmers, fishermen, small-scale business owners, transporters, and craftsmen/women,

yielding mean score values of 1.9, 1.7, 1.2, 1.4, and 1.5, respectively. There was a higher score in the perceptions of scenic beauty from civil servants and educators, as it had mean score values of 2.8 and 2.9, respectively. Supporting services, such as wild habitats, was highly important to fishermen and educators because they scored mean scores of 3.9 and 3.8, respectively. Soil formation attained a mean score value of 3.6 by educators. In contrast to small-scale business owners, craftsmen/women, and transporters, regulating and supporting services were highly perceived as mangrove ecosystem services, without a doubt, by those working in education-/research-related fields. This study found that no significant differences existed between the reliance on mangrove ecosystem services and household occupation, i.e., $\chi^2 = 8.015$ and $p = 0.1554$.

**Table 5.** Perception of mangrove ecosystem services by occupation in Chwaka, Charawe, Ukongoroni, Unguja Ukuu and Uzi wards in Unguja, Zanzibar: PS (provisioning services), RS (regulating services), CS (cultural services), and SS (supporting services).

| Mangrove Ecosystem Services | Occupation of Household Survey Participants | | | | | | |
|---|---|---|---|---|---|---|---|
| | Subsistence Farmers | Fishermen | Public Servants | Small-Scale Business | Transporters | Craftsmen/Women | Educators |
| | (*n* = 48) | (*n* = 36) | (*n* = 14) | (*n* = 28) | (*n* = 22) | (*n* = 13) | (*n* = 21) |
| **PS** | | | | | | | |
| Poles | 3.6 (0.5) | 3.4 (0.4) | 1.5 (0.6) | 3.5 (0.5) | 2.0 (0.2) | 3.4 (0.0) | 1.1 (0.2) |
| Firewood | 3.5 (0.7) | 3.7 (0.5) | 2.7 (0.4) | 3.1 (0.6) | 2.1 (0.4) | 3.1 (0.2) | 1.9 (0.0) |
| Medicines | 2.8 (0.6) | 3.2 (0.7) | 1.2 (0.2) | 2.3 (0.5) | 2.1 (0.4) | 2.4 (0.4) | 1.6 (0.3) |
| Fodder | 1.4 (0.4) | 1.5 (0.5) | 1.1 (0.3) | 1.1 (0.4) | 1.0 (0.5) | 1.5 (0.6) | 1.3 (0.7) |
| Fruit | 1.3 (0.2) | 1.6 (0.4) | 1.2 (0.2) | 1.1 (0.5) | 1.0 (0.6) | 1.1 (0.4) | 1.0 (0.4) |
| Picnic sites | 1.2 (0.2) | 1.1 (0.3) | 2.8 (0.6) | 1.9 (0.4) | 1.6 (0.2) | 1.7 (0.2) | 1.8 (0.0) |
| Honey | 1.1 (0.3) | 1.0 (0.4) | 1.0 (0.2) | 1.0 (0.3) | 1.0 (0.2) | 1.0 (0.4) | 1.0 (0.2) |
| **RS** | | | | | | | |
| Erosion control and sediment accretion | 2.9 (0.8) | 3.3 (0.6) | 3.0 (0.3) | 2.2 (0.5) | 1.7 (0.2) | 1.5 (0.4) | 3.8 (0.2) |
| Regulation of climate | 2.7 (0.5) | 2.6 (0.4) | 2.8 (0.6) | 2.6 (0.5) | 1.2 (0.4) | 1.6 (0.2) | 3.6 (0.4) |
| Attenuation of waves | 1.9 (0.3) | 3.2 (0.5) | 2.2 (0.0) | 1.4 (0.4) | 1.0 (0.2) | 1.3 (0.6) | 3.7 (0.6) |
| **CS** | | | | | | | |
| Scenic beauty | 1.9 (0.7) | 1.7 (0.5) | 2.8 (0.0) | 1.2 (0.4) | 1.4 (0.6) | 1.5 (0.2) | 2.9 (0.0) |
| Spiritual | 1.7 (0.5) | 1.8 (0.4) | 1.0 (0.0) | 1.1 (0.5) | 1.1 (0.2) | 1.7 (0.4) | 1.0 (0.2) |
| Ecotourism | 1.3 (0.3) | 1.9 (0.6) | 1.4 (0.2) | 1.4 (0.5) | 1.2 (0.6) | 1.6 (0.2) | 2.6 (0.4) |
| Education | 1.2 (0.2) | 1.3 (0.4) | 1.0 (0.0) | 1.3 (0.4) | 1.2 (0.2) | 1.2 (0.0) | 1.1 (0.6) |
| **SS** | | | | | | | |
| Wild habitat | 2.6 (0.5) | 3.9 (0.4) | 2.0 (0.6) | 1.1 (0.4) | 1.1 (0.4) | 1.9 (0.2) | 3.8 (0.4) |
| Soil formation | 1.3 (0.4) | 1.7 (0.5) | 1.1 (0.4) | 1.2 (0.2) | 1.0 (0.0) | 1.0 (0.2) | 3.6 (0.2) |

Note: means are represented by the values outside the parentheses, while the values inside the parentheses represent standard deviations. Likert scale: not important (1), least important (2), moderately important (3), most important (4), and not sure (5).

### 3.5. Factors Affecting Perception towards Mangrove Ecosystem Services

The findings from this study found that the perception and prioritization of respondents regarding mangrove ecosystem services were context-dependent and were determined by numerous factors such as engagement in mangrove conservation, age, education level, the socio-economic status of the local community, government policies related to the preservation of ecosystem services, the intensity and frequency of utilizing mangrove products, and the institutions responsible for managing them. During a focus group discussion in the Uzi ward, one participant reported that "the knowledge and experience on mangrove ecosystem service is based on my prolonged engagement in mangrove conservation and the periodic education on mangrove conservation I get from forest officers". The findings from the household surveys indicated that the most to the lowest influential factors that affected perceptions towards mangrove ecosystem services were education level, the intensity and frequency of utilizing mangrove products, engagement in mangrove conservation activities, the socio-economic status of the local community, and the institutions responsible for managing mangroves, as they recorded mean score values of 3.6, 2.7, 2.6, 2.4, and 2.3, respectively (Table 6). Two notable factors—the age of the respondents and government policies related to the conservation of ecosystem services—both attained a mean score of 1.2. Education level scored highest in the Chwaka, Ukongoroni/Charawe, and Unguja

Ukuu/Uzi wards. After education level, Chwaka rated the intensity and frequency of utilizing mangrove products as the second significant factor with a value of 2.9, compared to the Ukongoroni/Charawe ward with a value of 2.8 for engaging in mangrove conservation activities. The Unguja Ukuu/Uzi ward rated the second significant factor as the intensity and frequency of utilizing mangrove products with a value of 3.3.

**Table 6.** Factors affecting perceptions toward mangrove ecosystem services based on household survey in Chwaka, Charawe, Ukongoroni, Unguja Ukuu, and Uzi wards in Unguja, Zanzibar.

| Factors | Chwaka Ward | Ukongoroni/Charawe Wards | Unguja Ukuu/Uzi Wards | Overall |
|---|---|---|---|---|
| Engagement in mangrove conservation activities | 2.6 (0.9) | 2.8 (0.8) | 3.3 (0.7) | 2.6 (0.8) |
| Age of respondents | 1.1 (0.5) | 1.4 (0.4) | 1.1 (0.4) | 1.2 (0.4) |
| Education level | 3.4 (0.2) | 3.7 (0.4) | 3.8 (0.5) | 3.6 (0.4) |
| Socio-economic status of the local community | 2.4 (0.6) | 2.6 (0.5) | 2.2 (0.4) | 2.4 (0.5) |
| Government policies related to the conservation of ecosystem services | 1.2 (0.3) | 1.3 (0.9) | 1.1 (0.2) | 1.2 (0.5) |
| Intensity and frequency of utilizing mangrove products | 2.9 (0.7) | 2.6 (0.2) | 2.6 (0.3) | 2.7 (0.4) |
| Institutions responsible for managing mangroves | 2.6 (0.5) | 2.1 (0.6) | 2.3 (0.4) | 2.3 (0.5) |

Note: means are represented by the values outside the parentheses, while the values inside the parentheses represent standard deviations. Likert scale: (1) strongly disagree, (2) disagree, (3) neither agree or disagree, (4) agree, (5) and strongly agree.

### 3.6. Threats to Mangrove Ecosystem Services

About 76% (*n* = 31) of the participants in the key informant interviews claimed that mangroves were degraded during the extraction of ecosystem services; the remaining 24% opined that severe degradation occurred during the extraction of mangrove-related resources. In the Unguja Ukuu ward, 74.5% of the participants claimed that they had witnessed illegal harvesting of mangrove products in the study area more than once. A participant in the focus group discussions stated that although it is unlawful to harvest mangrove products without a permit, locals do so covertly, endangering the regeneration of mangrove ecosystems. The participants in the focus group discussions and key informant interviews agreed that mangroves were more degraded in unprotected mangrove forests. The participants in the focus group discussions agreed that the amount of mangrove ecosystem services extracted in the study area was far more than the regeneration amount. According to the respondents in the focus group discussions, the rate at which mangrove ecosystem services are harvested, especially in Unguja Ukuu, is worrying; local loggers work in cahoots with corrupt community leaders and government officials.

### 3.7. Factors Affecting the Supply of Mangrove Ecosystem Services

The factors affecting mangrove ecosystem services in the Chwaka and Menai Bays are shown in Table 7. During the focus group discussions in the Chwaka ward, one participant reported that "Mangroves' natural beauty has been damaged, and fish breeding grounds have been severely affected". An academic staff member at The State University of Zanzibar also stated that "Local communities around the bays are poor, and such areas have rapid population growth. Also, the lack of alternative livelihood opportunities drives them to rely on illegal mangrove harvesting. However, the forest department lacks adequately trained personnel due to insufficient funding from the government". The strongest drivers identified by the household survey were lack of alternative livelihoods and unresponsive or faulty institutional structures in terms of forestry management, which achieved overall mean score values of 3.5 (Table 7). Insufficient resources to fund institutions and hire related expertise in managing forestry resources attained mean scores of 3.4, 3.3, and 3.1

for the Chwaka, Ukongoroni/Charawe, and Unguja Ukuu/Uzi wards, respectively. The unclear/faulty/discriminatory land tenure system also showed a relatively strong influence, with mean score values of 3.4, 2.9, and 3.2 for the Chwaka, Ukongoroni/Charawe, and Unguja Ukuu/Uzi wards, respectively. Flawed/unclear/opaque/poorly formulated forestry policy and legal framework attained an overall mean score of 1.2; Chwaka, Ukongoroni/Charawe, and Unguja ukuu/Uzi wards attained mean score values of 1.2, 1.0, and 1.3, respectively.

**Table 7.** Factors affecting mangrove ecosystem services based on household survey in Chwaka, Charawe, Ukongoroni, Unguja Ukuu, and Uzi wards in Unguja, Zanzibar.

| Drivers of Changes in Mangrove Ecosystem Services | Chwaka Ward | Ukongoroni/Charawe Wards | Unguja Ukuu/Uzi Wards | Overall |
|---|---|---|---|---|
| Lack of alternative livelihood | 3.5 (0.6) | 3.4 (0.4) | 3.6 (0.6) | 3.5 (0.5) |
| Insufficient resources to fund institutions and hiring of adequate personnel to manage forestry resources | 3.4 (0.5) | 3.3 (0.9) | 3.1 (1.1) | 3.3 (0.8) |
| Unresponsive/faulty institutional structure in forestry management | 3.6 (0.6) | 3.7 (0.6) | 3.3 (0.5) | 3.5 (0.6) |
| Population growth | 2.4 (0.8) | 2.9 (0.2) | 2.8 (0.6) | 2.7 (0.5) |
| Climate change | 2.1 (0.4) | 2.8 (0.6) | 2.5 (0.9) | 2.5 (0.6) |
| Unresponsive/faulty government forestry policy | 2.0 (0.5) | 2.5 (0.7) | 2.1 (0.5) | 2.2 (0.6) |
| Agricultural development | 1.3 (0.4) | 1.3 (0.6) | 1.8 (0.4) | 1.5 (0.5) |
| Expanded aquaculture | 2.0 (0.4) | 1.4 (0.2) | 1.1 (0.8) | 1.5 (0.5) |
| Unchecked/uncontrolled coastal development | 2.9 (0.3) | 1.3 (0.7) | 1.4 (1.0) | 1.9 (0.7) |
| Underfunded judicial system | 1.6 (0.6) | 1.7 (0.9) | 1.3 (0.3) | 1.5 (0.6) |
| Weak government institutions | 1.9 (0.6) | 2.3 (0.2) | 2.1 (0.4) | 2.1 (0.3) |
| Flawed/unclear/opaque/poorly formulated forestry policy and legal framework | 1.2 (0.4) | 1.0 (0.5) | 1.3 (0.8) | 1.2 (0.7) |
| Lack of policy/laws specifically for mangroves | 2.2 (0.5) | 2.4 (0.2) | 2.1 (0.6) | 2.2 (0.2) |
| Local communities feel they are entitled to ownership of and access to mangrove ecosystem services | 2.6 (0.4) | 2.4 (0.8) | 2.6 (0.2) | 2.5 (0.7) |
| Lack of expertise and adequate staff in management of mangroves at local level | 1.4 (0.5) | 1.7 (0.3) | 1.9 (1.0) | 1.7 (0.6) |
| Poor public participation in design and decision-making involving mangrove ecosystem services | 2.6 (0.8) | 3.0 (0.5) | 2.3 (0.5) | 2.6 (0.6) |
| Unclear/faulty/discriminatory land tenure system | 3.4 (0.9) | 2.9 (0.6) | 3.2 (0.6) | 3.2 (0.7) |
| Corruption | 2.8 (0.6) | 2.5 (0.4) | 2.7 (0.5) | 2.7 (0.5) |

Note: means are represented by the values outside the parentheses, while the values inside the parentheses represent standard deviations. Likert scale: (1) not important, (2) least important, (3) second most important, and (4) most important.

## 4. Discussion

### 4.1. Awareness of Mangrove Ecosystem Services in Chwaka and Menai Bays

This study used focus group discussions, household surveys, and key informant interviews to assess the perception and contribution of mangrove ecosystem services to the well-being of communities in the Chwaka and Menai Bays. People's perceptions of ecosystem goods and services are essential to understanding the interaction between people and their environment [52]. High levels of community awareness influence people's perceptions of environmental benefits and make it easier to recognize how ecosystems support their livelihoods and well-being [53]. Provisioning services were the most-identified mangrove ecosystem services, which was in line with the findings of Oteros-Rozas et al. [54]; provision-

ing services were mostly identified because of their high market value. Cultural services were second. Also, the results of this study showed that because forest-adjacent communities are sometimes involved in the conservation and restoration of mangrove ecosystems, they seemed to be aware of regulating and supporting services. Ruslan et al. [20] reported that locals who participate in mangrove conservation activities are more likely to appreciate the provisioning and supporting services of mangroves.

*4.2. Perception and Importance of Mangrove Ecosystem Services to the Livelihoods and Well-being of the Communities in Chwaka and Menai Bays*

In sustaining local livelihoods, firewood and poles were the highest-rated provisioning services by the focus group discussions, while the key informant interviews highly rated regulating services. Due to their accessibility, mangrove poles are used for building houses and making furniture, which is in line with Mensah et al. [55], who found that when forest products and services are accessible, local communities recognize them. Most locals rely on mangroves for fuel; Makonese et al. [56] noted that forest-adjacent communities use firewood for cooking since it is more readily available and less expensive. The Unguja Ukuu and Uzi wards had higher mean scores for supporting services, such as erosion control and sediment accretion, climate, and attenuation of waves, and regulating services such as wild habitats and soil formation. Recently, residents in both wards have been involved in workshops on the conservation and restoration of mangroves. Educating community members on ecosystem protection makes them more knowledgeable about supporting and regulating services [29]. Local communities explore the benefits of ecosystem services as awareness of the regulating and supporting services of mangroves grows [53]. Because it is challenging to keep bees in mangrove areas, the respondents placed minimal weight on honey production. Lee et al. [26] reported that it is challenging to keep bees for honey in forests where there is considerable mangrove cutting. Fruits and fodder were among the provisioning services that were recognized by a small number of the respondents. According to Tanner et al. [23] and Joshi and Negi [57], farmers only graze livestock in mangrove wetlands whenever there is a shortage of fodder, especially during dry seasons. Farmers in the study area asserted that there was little usage of mangrove feed because animals frequently graze on coral rags, grasses, and bushes. Traditional medicine was mentioned by several respondents and was placed high among the provisioning services. Forest-adjacent communities frequently rely on conventional remedies over contemporary healthcare facilities [58].

The findings demonstrated that the Unguja Ukuu and Uzi wards valued the significance of regulating services, which include protecting shorelines and controlling climate. The household survey findings showed that local communities, particularly in the Unguja Ukuu and Chwaka wards, valued the attenuation of waves since they had severally experienced coastal flooding in the past, which caused significant marine erosion. The same conclusions were made by Damastuti and de Groot [59], who demonstrated that the frequency of coastal threats, such as tsunamis and storms, makes coastal people appreciate mangrove ecosystem services, such as coastal protection. This is consistent with Joshi and Negi [57]; dense mangroves shield coastal areas from powerful storms and waves.

The medium scores for cultural services were for spiritual beliefs, which may be attributed to the transition of coastal communities from traditional spiritual beliefs to modern religious beliefs in Islam and Christianity [60]. Provisioning services, such as poles, firewood, and medicine, were given the highest scores by the Chwaka, Ukongoroni, Charawe, Unguja Ukuu, and Uzi wards, which was in contrast to honey, fruit, picnic sites, and fodder, which were given lower scores. The Chwaka, Ukongoroni, and Charawe wards gave cultural services, such as places for spiritual activities and scenic beauty, higher ratings than regulating services, which was likely due to their direct advantages that positively impact the communities' means of livelihood. Cultural services, e.g., education and ecotourism, also received high scores. These findings were similar to Lee et al. [26], who demonstrated that coastal communities have different perceptions of ecosystem services in their localities

because of socioeconomic status. Fishing is one of the most vital sources of income on Zanzibar Island; therefore, the Unguja Ukuu and Uzi wards highly valued habitat-related services, but the Chwaka, Ukongoroni, and Charawe wards awarded wild habitats the lowest marks since they did not see the importance of mangroves as habitat grounds.

This study found that mangrove ecosystem services could be related to the residents' livelihood sources. For instance, small-scale farmers and fishermen/women highly ranked the importance of providing services like poles, firewood, and medicine. In contrast, educated respondents and those in the teaching and research professions gave the highest priority to regulating services such as soil erosion control and sediment accretion, climate regulation, and wave attenuation [13]. Ruiz-Frau et al. [61] found that respondents' propensity to recognize ecosystem services depends on the types of employment they are engaged in. According to He et al. [52], small-scale farmers and fishers strongly value provisioning and cultural services since they interact with and use them directly.

The results of this study showed that the respondents' perceptions of mangrove ecosystem services depended on the context and were influenced by a wide range of variables. Examples include the involvement of local communities in mangrove conservation, socioeconomic status, and government policies relating to the preservation of ecosystem services. Others include accessibility to mangrove ecosystem services and the institutions tasked with managing mangroves. According to Purida et al. [35], various social and political elements can affect how a local population perceives ecosystem services. The results of this study are consistent with those of the Millennium Ecosystem Assessment [8] and Marlianingrum et al. [31], which revealed that respondents' education levels and involvement in mangrove protection are the two most important factors influencing how people perceive ecosystem services. The socioeconomic level of the local population, interaction with the environment, age, education, and interaction with the ecosystem have significant roles in shaping how local people perceive ecosystem services [23].

### 4.3. Extraction of Mangrove Ecosystem Services and Destruction of Mangrove Forests in Zanzibar

Due to rapid population growth and changes in land use, the mangroves in the Chwaka and Menai Bays have experienced a significant rate of deterioration in coverage and quality [44]. The respondents from the five wards agreed on the extensive illegal harvesting of mangroves, reducing mangrove ecosystem services. These findings are supported by Msangameno et al. [62], who stated that the loss in mangrove coverage in these bays has resulted in low ecological benefits to the coastal people in Zanzibar. More significantly, this problem has endangered the culture of mangrove-dependent Zanzibar people [42]. Furthermore, Zanzibar's mangrove ecosystem benefits are grossly undervalued [12], since most ecosystem services are illegally traded [31]. The respondents knew of the adverse impacts, particularly coastal erosion, along the coastal shore as mangroves decline. The severe coastal erosion in the Zanzibar Islands has worsened due to ongoing massive coastal deforestation [63]. In particular, it has been noted that considerable erosion has been detected along the 113.5 km of sandy beaches in Unguja, which has been highly attributed to deforestation and tourism development [64]. Since the 1980s, Zanzibar's unexpected increase in tourism and related development has increased the demand for forest products [63]. This trend has continued unabated up to date.

### 4.4. Drivers of Changes in Mangrove Ecosystem Services in Zanzibar

This study also documented several drivers of mangrove change in the Chwaka and Menai Bays. The household survey identified a lack of alternative livelihoods, insufficient resources to fund institutions and acquire adequate staff to manage forestry resources, unresponsive or faulty institutional structures in forestry management, and the unclear and/or discriminatory land tenure system in Zanzibar. The findings in this study agreed with Ruslan et al. [20] in that a lack of alternative livelihoods, a lack of qualified labor force in mangrove conservation and management, ineffective management, and population pressure are the main drivers of mangrove forest change and ecosystem services.

Kukkonen et al. [38] found that mangrove forests in the Chwaka and Menai Bays have experienced illegal harvesting of their products. In general, illicit, unsustainable resource exploitation severely threatens numerous eastern African coastal resources [47,65]. Also, the respondents reported climatic change as a threat to the ecosystem services provided by mangrove ecosystems [13]. Generally, changes in mangrove ecosystem services considerably impact ecosystems and the quality of human life.

*4.5. Limitation of the Study*

This study experienced the limitations of qualitative research regarding subjectivity and researcher bias. The sampling design, the interview guide, deciding which interviewees to contact, the coding process, systematizing the study results, and interpreting them all involve some bias [66]. The snowballing sampling approach used to sample the interviewees in this study is biased because interviewees may share a similar perception [67]. The discourse analysis used in this study can be subjective, and its findings are often specific to a particular context, limiting the findings' generalizability. As much as the study aimed to achieve a more heterogeneous perception of mangrove ecosystem services, some interviewees were not directly responsible for the sustainable management of mangroves. Also, convincing the local communities to participate in the study took work, as they were skeptical and wondered how it would benefit them. They complained that researchers come to them asking for information but have not seen any benefit.

Generally, the smaller the population, the larger the sampling ratio required; for populations below 1000, a minimum ratio of 30 percent, i.e., 300 individuals, is needed to ensure the sample's representativeness [68]. For larger populations, such as 10,000, a relatively small minimum ratio of 10 percent, i.e., 1000 individuals, is required [68]. This study's comparatively small sample size was due to the fact that the target groups were small and hard to access; navigating through the dense mangroves using a canoe was challenging and time-consuming.

Additionally, navigating through the dense mangrove trees using a canoe was challenging and time-consuming. Also, some respondents' comprehension of the questionnaires took a lot of work; therefore, collecting the data took time and effort. The acquisition of a research permit from the Department of Forestry took longer than anticipated and affected every other aspect of the study.

**5. Conclusions**

This study aimed to assess how people in the Chwaka and Menai Bays perceive ecosystem services and the contribution of mangrove ecosystem services to their well-being. The respondents perceived the ecosystem services differently according to focus group discussions, key informant interviews, and household surveys. Perceptions about mangrove ecosystem services are an essential communication and decision-support tool for policies on managing and conserving mangrove ecosystems. Generally, understanding preferences for mangrove ecosystem services can strengthen the links between local stakeholders and conservation actors for the sustainable management of mangroves. Overall, provisioning services were the most commonly recognized as supporting local livelihoods due to their direct value. The comparatively lower rating of regulating, cultural, and supporting services highlights the need for current management initiatives to increase awareness about mangrove forests' value, significance, and multifaceted functionality. Such initiatives can motivate local communities and management institutions to relook at the various benefits of mangroves; this could contribute to more sustainable use/management of mangrove ecosystem services to benefit posterity.

The value of ecosystem services cannot be ignored. There is still a need for further research into mangrove ecosystem services. Future research will aim to study the economic value of mangrove ecosystem services in Zanzibar.

**Author Contributions:** Conceptualization, M.K.M.; methodology, M.K.M.; software, M.K.M.; validation, M.K.M. and E.A.; formal analysis, M.K.M.; investigation, M.K.M.; resources, M.K.M.; data curation, M.K.M.; writing—original draft preparation, M.K.M.; writing—review and editing, M.K.M., E.A. and C.M.J.; visualization, M.K.M.; supervision, E.A. and C.M.J. All authors have read and agreed to the published version of the manuscript.

**Funding:** This research received no external funding.

**Data Availability Statement:** The data presented in this study are available on request from the corresponding author. The data are not publicly available due to privacy.

**Conflicts of Interest:** The authors declare no conflict of interest.

## Appendix A

My name is ________________________. I am conducting an academic survey to collect information about the awareness among the communities of Chwaka and Menai Bays of the benefits of mangrove forests. Therefore, your assistance filling out this questionnaire will go a long way in my academic pursuit and conservation of mangrove forests. Your responses may be used to influence the policy direction of mangroves. Your participation is voluntary and confidential. Do not indicate your name or any other form of identification.

Record Number: ________________

Demographic and socio-economic information of respondent

| Date: | Time: |
|---|---|
| District: | Ward: |

| 1. Household composition | |
|---|---|
| Religion: (1) Muslim (2) Christian | Marital Status: (1) Not married (2) Married (3) Widowed (4) Divorced |
| Gender: (1) Male (2) Female | Education: (1) No formal education (2) Primary (3) Secondary (4) Tertiary |
| Age: (1) Under 20 (2) 20–39 (3) 40–49 (4) 50–59 (5) 60–69 (6) 70+ | Household size: (1) 1–5 (2) 6–10 (3) 10+ |
| Main occupation: (1) Crafts (2) Bee keeping (3) Transporters (4) Farming (5) Fishing (6) Small business (7) Salaried employee (8) Wage laborer 10. Other (specify)________________ | |

2. For how long have you in this ward: (1) <5 year (2) 5–10 years (3) 10–15 years (4) 15–20 years (5) >20 years

3. Household income per month: (1) <100,000 (2) 100,000–200,000 (3) >200,000

4. Source of cooking energy: (1) Electricity/gas (2) Firewood/charcoal (3) Other (specify) ____________

5. Distance from the village to the forest: ________km

Mangrove ecosystem services

6. How much do you agree that mangroves give benefits to the community?
(1) Strongly disagree (2) Disagree (3) Neither agree or disagree (4) Agree (5) Strongly agree

7. On a scale of 1–5, indicate your level of awareness of mangrove benefits in your area: not aware (1), slightly aware (2), somewhat aware (3), moderately aware (4), extremely aware (5).

| Benefits | 1 | 2 | 3 | 4 | 5 |
|---|---|---|---|---|---|
| Poles | | | | | |
| Firewood | | | | | |
| Medicine | | | | | |
| Honey | | | | | |
| Fruits | | | | | |
| Picnic sites | | | | | |
| Fodder | | | | | |
| Control of soil erosion | | | | | |
| Regulation of climate | | | | | |
| Attenuation of waves | | | | | |
| Wild habitat | | | | | |
| Soil formation | | | | | |
| Ecotourism | | | | | |
| Spiritual activities | | | | | |
| Educational purposes | | | | | |
| Scenic viewing | | | | | |

8. On a scale of 1–5, indicate the level of importance of each mangrove benefit to the community: not important (1), least important (2), moderately important (3), most important (4), not sure (5).

| Benefits | 1 | 2 | 3 | 4 | 5 |
|---|---|---|---|---|---|
| Poles | | | | | |
| Firewood | | | | | |
| Medicine | | | | | |
| Honey | | | | | |
| Fruits | | | | | |
| Picnic sites | | | | | |
| Fodder | | | | | |
| Control of soil erosion | | | | | |
| Regulation of climate | | | | | |
| Attenuation of waves | | | | | |
| Wild habitat | | | | | |
| Soil formation | | | | | |
| Ecotourism | | | | | |
| Spiritual activities | | | | | |
| Educational purposes | | | | | |
| Scenic viewing | | | | | |

9. Which mangrove ecosystem services listed in 8 above have you benefited from (list them in order of preference)?

| | |
|---|---|
| 1. | |
| 2. | |
| 3. | |
| 4. | |
| 5. | |
| 6. | |
| 7. | |
| 8. | |
| 9. | |
| 10. | |
| 11. | |
| 12. | |
| 13. | |
| 14. | |
| 15. | |
| 16. | |

10. Do you agree or disagree that the following factors have contributed to how people perceive mangrove ecosystem services? (1) Strongly disagree, (2) Disagree, (3) Neither agree/disagree, (4) Agree, (5) Strongly agree

| Factors | 1 | 2 | 3 | 4 | 5 |
|---|---|---|---|---|---|
| Engagement in mangrove conservation activities | | | | | |
| Age of respondents | | | | | |
| Education level | | | | | |
| Socio-economic status of the local community | | | | | |
| Government policies related to the conservation of ecosystem services | | | | | |
| Intensity and frequency of utilizing mangrove products | | | | | |
| Institutions responsible for managing mangroves | | | | | |

11. Do you agree that the following factors have affected the benefits from mangrove forests in the ward? (1) Strongly disagree, (2) Disagree, (3) Neither agree or disagree, (4) Agree, (5) Strongly agree

| Factors | 1 | 2 | 3 | 4 | 5 |
|---|---|---|---|---|---|
| Insufficient resources to fund institutions & hiring of adequate personnel to manage forestry resources | | | | | |
| Unresponsive/faulty institutional structure in forestry management | | | | | |
| Population growth | | | | | |

| |
|---|
| Climate change |
| Unresponsive/faulty government forestry policy |
| Agricultural development |
| Expanded aquaculture |
| Unchecked/uncontrolled coastal development |
| Underfunded judicial system |
| Weak government institutions |
| Flawed/Unclear/opaque/poorly formulated forestry policy and legal framework |
| Lack of policy/laws specifically for mangroves |
| Local communities feel they are entitled to ownership and access to mangrove ecosystem services |
| Lack of expertise and adequate staff in management of mangroves at local level |
| Poor public participation in design & decision-making involving mangrove ecosystem services |
| Unclear/faulty/discriminatory land tenure system |
| Corruption |

12. Rate change in mangrove ecosystem services in the ward compared to 10 years ago.
(1) Substantial decrease (2) decreased (3) Same (4) Increased (5) Substantial increase (6) Not sure

13. Explain why change has or has not occurred.
-------------------------------------------------------------------------------------------------------
-------------------------------------------------------------------------------------------------------
-------------------------------------------------------------------------------------------------------
-------------------------------------------------------------------------------------------------------
-------------------------------------------------------------------------------------------------------
-----------------------------------

14. What do you think will be the main reasons for changes in the benefits accrued from mangrove forests over the next 10 years?
-------------------------------------------------------------------------------------------------------
-------------------------------------------------------------------------------------------------------
-------------------------------------------------------------------------------------------------------
-------------------------------------------------------------------------------------------------------
-------------------------------------------------------------------------------------------------------
-----------------------------------

15. Do you agree or disagree that it is important to protect mangrove forests for posterity?
(1) Strongly disagree (2) Disagree (3) Neither agree/disagree (4) Agree (5) Strongly agree

16. To what extent do you agree or disagree that if mangrove forests are conserved, there will be more mangrove ecosystem services?

(1) Strongly disagree (2) Disagree (3) Neither agree/disagree (4) Agree (5) Strongly agree
Please provide any additional comments here below:
------------------------------------------------------------------------------------------------------------
------------------------------------------------------------------------------------------------------------
------------------------------------------------------------------------------------------------------------
------------------------------------------------------------------------------------------------------------
------------------------------------------------------------------------------------------------------------
----------------------------------

Thanks for your time and participation.

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
