# Peer review of "Assessing the Perception and Contribution of Mangrove Ecosystem Services to the Well-Being of Coastal Communities of Chwaka and Menai Bays, Zanzibar"

_resources, doi:10.3390/resources13010007_

Round 1
Reviewer 1 Report
Comments and Suggestions for Authors
Please see the attached file.
Actually, I liked the subject and the method used in this study. Such research is imperative for all policy-makers and land managers.
some comments need to be considered for better understanding:
for example: I think the authors use ample abbreviations or acronyms. It is better to abbreviate several non-essential terms!
The literature review needs a significant revision. In its current form, it is not acceptable.
What are the limitations of the study? mention them in the discussion or conclusion!

Comments on the Quality of English LanguageModerate editing of the English language is required.
Reviewer 2 Report
Comments and Suggestions for Authors
This paper is overall well-structured and the writing quality is fine. Research questions and purposes of this paper are clearly stated. However, before consideration for publication, I would like to recommend some minor corrections for the improvement of this study.
1.The abstract should be re written. The abstract should state briefly the purpose of the research, the principal results and major conclusions. The current version is too generic. Summary does not require too many formulas and numbers.
2. Literature review needs to be done more systematically.
3.The paper should provide a detailed explanation of the relevant concepts and connotations such as ecosystem services.
4. Limitations of the study are missing.
Reviewer 3 Report
Comments and Suggestions for Authors
This is an informative paper regarding the perception of mangrove ecosystem services in Zanzibar. As the authors mention the aim of their research was to assess the perception and value of MES to the people living in Chwaka and Menai Bay areas, in three groups of wards. The four research questions are Which MES exist in Chwaka and Menai Bays? , What is the significance of the MES to the well-being of the communities in the Chwaka and Menai Bay areas? , Does the extraction of MES contribute to the destruction of mangrove forests? and What are the factors of supply of MES, and how have they affected the livelihoods of the bay-adjacent communities?
Still, I missed a hypothesis, What were the authors expecting to find?. The conjunction of statistical and qualitative data is interesting, but readers need to understand what was prior to the research assumptions. Answering the four questions only is too descriptive and the discussion and conclusion is weak.
There assumptions like "The respondents were believed to be knowledgeable in MES". How can that be? how belief is the source of sampling?
I suggest to add the questionnaire as an appendix.
Arrange table 3 by joining the same ES, this would give order and would help to understand Figure 3 as well.
Table 4 text is far away from the table itself.
In section 3.2 species are not in italics
I cannot see how the authors conclude that by having a more knowledgeable population about ES the mangrove deterioration would be controlled. I suggest checking the conclusion section and rethinking the statements.
Reviewer 4 Report
Comments and Suggestions for Authors
This study tries to evaluate mangrove ecosystem services in Zanzibar. The study identifies the various ecosystem services provided by mangroves and highlights their crucial role in the livelihoods of local communities. The authors have gathered information from government officers and academicians from universities in Zanzibar who hold expertise in mangrove ecosystems. The following topics were chosen for the semi-structured interviews: the concept of ecosystem services, ecosystem services of mangroves, categorization of mangrove ecosystem services, and perceptions toward mangrove ecosystem services. Understanding the value of mangrove ecosystem services is essential for promoting human well-being, sustainability, and distributive justice.
In general, the article is well-crafted, and the results are promising and essential, contributing to the promotion of co-benefits for the local communities relying on mangroves in the face of conservation and restoration projects. Increasingly, the role of ES provided by mangroves has been recognized, not only for climate change mitigation but also for enhancing the lives of local communities. However, the article requires extensive English revision to enhance sentence clarity, readability, and the use of scientific language. Another point is that the study's sample size was relatively small, and the findings may not be generalizable to other regions which could improve the study's relevance. Additionally, the authors could try to include an economic valuation of the ecosystem services provided by mangroves, which could have provided more insights into their value.
Abstract
There's a need for a more detailed description of the analysis method. It only mentions the use of interviews. Please provide additional details about the evaluation methods.
Introduction
The introduction could be improved by reducing the number of sentences used to contextualize the article's purposes. In this case, a shorter introduction defining the research gaps and outlining the study's objectives and hypotheses would be more effective.
Materials and Methods
It's not clear whether the authors used an Informed Consent Form for the interviews. Could you please clarify this issue?
Results
In my opinion, the results also suffer from an excess of unnecessary information. There are overlaps between results and text that could be moved to the Materials and Methods section. I believe this part needs extensive revision. Additionally, the results section contains some misplaced captions and missing tables.
Discussion
Overall, the discussion section provides a comprehensive analysis of the study's results and their implications for the management and conservation of mangroves in Zanzibar.

Comments on the Quality of English LanguageThe article requires extensive English revision to enhance sentence clarity, readability, and the use of scientific language.
Round 2
Reviewer 4 Report
Comments and Suggestions for Authors
I am pleased to review once again the article authored by Mohamed et al. titled "Assessing the perception and contribution of mangrove ecosystem services to the well-being of coastal communities of Chwaka and Menai Bays, Zanzibar." In this revised version, the authors have diligently addressed all the points raised in the previous manuscript. Consequently, the new version has significantly improved and is nearing publication. However, I believe that the article still requires a more detailed review of its English, employing a more scientific language. Below are some sentences that do not convey a clear idea to the reader:
1. "offered by mangroves": Mangroves do not offer; they provide ecosystem services.
2. "Ecosystem benefits are vital to our survival, providing us with clean air, freshwater, and fertile soil, among other things." Please specify "other things." It is advisable to maintain clarity in sentences and avoid using informal language in a scientific article.
3. "ecosystem services mangroves": The phrasing of this sentence confuses the concept of ecosystem services. It suggests that there is a class of mangrove ecosystem services within the concept, which is not accurate.
4. "The management of ecosystem services": While I understand that this expression has been used, how is it possible to manage the service? I understand that it is possible to manage the ecosystem to perform its functions to provide ecosystem services.
5. "Menai Bay is composed of c. 988 acres of mangroves." What does "c. 988 acres" mean? Is it 988 hectares (ha)?
In addition to the issues of scientific writing, the authors have not provided a clear justification for the relatively small sample size in the study.
Comments on the Quality of English LanguageThe article still requires a more detailed review of its English, employing a more scientific language.
